

# Why is ozone in South Korea and the Seoul Metropolitan Area so high and increasing?

Nadia K. Colombi[1*], Daniel J. Jacob[1], Laura Hyesung Yang[2], Shixian Zhai[2], Viral Shah[3,4], Stuart
K. Grange[5], Robert M. Yantosca[2], Soontae Kim[6], and Hong Liao[7]

[1]Harvard University, Department of Earth and Planetary Science, Cambridge, MA 02138, USA

[2]Harvard University, John A. Paulson School of Engineering and Applied Sciences, Cambridge, MA 02138, USA

[3]NASA Global Modeling and Assimilation Office, Goddard Space Flight Center, Greenbelt, MD 20771, USA

[4]Science Systems and Applications, Inc., Lanham, MD 20706, USA

[5]Empa, Swiss Federal Laboratories for Materials Science and Technology, Überlandstrasse 129 8600 Dübendorf, Switzerland

[6]Ajou University, Department of Environmental and Safety Engineering, Suwon, Gyeonggi 16499, Republic of Korea

[7]Jiangsu Key Laboratory of Atmospheric Environment Monitoring and Pollution Control, Jiangsu Collaborative Innovation Center of Atmospheric Environment and Equipment Technology, School of Environmental Science and Engineering, Nanjing University of Information Science & Technology, Nanjing 210044, China

*Correspondence to: Nadia Colombi (ncolombi@g.harvard.edu)

**Abstract.** Surface ozone pollution in South Korea has increased over the past two decades, despite efforts to decrease emissions, and is pervasively in exceedance of the maximum daily 8-hr average (MDA8) standard of 60 ppb. Here, we investigate the 2015-2019 trends in surface ozone and $NO_2$ concentrations over South Korea and the Seoul Metropolitan Area (SMA), focusing on the 90th percentile MDA8 ozone as an air quality metric. We use a random forest algorithm to remove the effect of meteorological variability on the 2015-2019 trends and find an emission-driven ozone increase of up to 1.5 ppb $a^{-1}$ in April-May while $NO_2$ decreases by 22%. GEOS-Chem model simulations including recent chemical updates can successfully simulate surface ozone over South Korea and China as well as the very high free tropospheric ozone observed above 2 km altitude (mean 75 ppb in May-June), and can reproduce the observed 2015-2019 emission-driven ozone trend over the SMA including its seasonality. Further investigation of the model trend for May, when meteorology-corrected ozone and its increase are the highest, reveals that a decrease in South Korea $NO_x$ emissions is the main driver for the SMA ozone increase. Although this result implies that decreasing volatile organic compound (VOC) emissions is necessary to decrease ozone, we find that SMA ozone would still remain above 80 ppb even if all anthropogenic emissions in South Korea were shut off. China contributes only 8 ppb to this elevated South Korea background and ship emissions contribute only a few ppb. Zeroing out all anthropogenic emissions in East Asia in the model indicates a remarkably high external background



of 56 ppb, consistent with the high concentrations observed in the free troposphere, implying that the air quality standard in South Korea is not practically achievable unless this background external to East Asia can be decreased.

**1 Introduction**


Surface ozone is a severe air quality problem in South Korea and has become steadily worse over the past two decades (Yeo and Kim, 2021; Kim et al., 2021). Ozone often exceeds 90 ppb in the Seoul Metropolitan Area (SMA) where 50% of South Korea's population is located (Miyazaki et al., 2018). In 2015, Phase 2 of the Seoul Metropolitan Air Quality Control Master Plan established a standard of 60 ppb for the maximum daily 8-hour

average (MDA8) ozone concentration (MOE, 2016). However, no monitoring sites have been compliant with this standard in recent years and ozone has continued to increase (NIER, 2020). Improved understanding of the causes of elevated ozone in South Korea is crucial for developing effective emission control strategies.

Ozone is produced in the troposphere by photochemical oxidation of volatile organic compounds (VOCs) in the

presence of nitrogen oxides ($NO_x \equiv NO + NO_2$). Both VOCs and $NO_x$ have large anthropogenic sources from combustion, and VOCs also have fugitive industrial and residential, as well as biogenic sources. Effectively reducing ozone concentrations requires knowledge of whether ozone production is $NO_x$- or VOC-limited. In the $NO_x$-limited regime, decreasing $NO_x$ emissions decreases ozone while decreasing VOC emissions has little effect. In the VOC-limited regime, when $NO_x$ concentrations are very high, decreasing $NO_x$ emissions drives an increase in

ozone while decreasing VOC emissions decreases ozone (Sillman et al., 1990). The Clean Air Policy Support System (CAPSS) bottom-up emission inventory in South Korea reports emission declines of 26% for $NO_x$ and 25% for VOCs in Seoul for the 2000-2019 period (https://www.air.go.kr/eng/capss/emission/sido.do?menuId=100). Using satellite and surface observations of $NO_2$, Seo et al. (2021) found that $NO_x$ emissions declined in Seoul by 30% during the 2015-2019 period, and Bae et al. (2021) found a 18% decrease for the 2015-2018 period. On the

other hand, Bauwens et al. (2022) found an increase in satellite-observed HCHO columns over South Korea by 1-2% $a^{-1}$ for the 2005-2019 period, which does not support a decrease in VOC emissions.

Ozone concentrations over South Korea depend not only on domestic emissions but also on the background from external sources. The KORUS-AQ aircraft campaign in May-June 2016 found free tropospheric concentrations

above 2 km altitude frequently exceeding 80 ppb (Miyazaki et al., 2018; Gaudel et al., 2018, Gaubert et al, 2020; Crawford et al., 2021), which would affect surface ozone through subsidence. An obvious source of background ozone is China, where ozone is very high and increasing (K. Li et al., 2019, 2021), and would be transported to South Korea by westerly winds (Cuesta et al., 2018). But other background sources could also contribute. Lam and Cheung (2022) found that strong transport from the stratosphere can enhance springtime surface ozone by up to 8

ppb in East Asia. J. Li et al. (2016) estimated from a global model that long-range transport from outside East Asia could contribute 50-80% to annual surface ozone in the Korean peninsula. Wang et al. (2022) found an increase of





free tropospheric ozone over East Asia from aircraft and ozonesonde data of 3.8 to 6.7 ppb per decade over the 1995-2017 period.

The high and increasing ozone over South Korea could thus reflect a combination of decreasing $NO_x$ emissions and/or increasing VOC emissions under VOC-limited conditions for ozone production (Jung et al, 2018), as well as high and increasing background ozone. Here we aim to better understand the factors controlling ozone and its increase in the SMA and more broadly over South Korea during the 2015-2019 period. We use a random forest (RF) method (Grange et al., 2018) to correct for the role of meteorology in driving the 2015-2019 ozone trend in the

SMA, and show that meteorology-corrected ozone is highest in May-June and increases the fastest in April-May, while showing no significant trend in July-August. We find that the GEOS-Chem chemical transport model can successfully capture the magnitude and trends of ozone concentrations, including their seasonality, and we use the model to quantify the importance of domestic and different background contributions in driving elevated ozone and its increase over South Korea.


**2 AirKorea data and trends, 2015-2019**

We use ozone and $NO_2$ concentrations measured hourly by the AirKorea national air quality network of the South Korea Ministry of Environment (http://www.airkorea.or.kr/web). There are 255 sites in South Korea covering the

2015 to 2019 period including 79 sites in the SMA defined here as the rectilinear domain (126.7°E–127.3°E, 37.2°N–37.8°N). Figure 1 shows the maximum monthly 90[th] percentile MDA8 ozone at the ensemble of AirKorea sites for each year from 2015 to 2019. Ozone rises steadily over that period except for a dip in 2018, reaching 96 ppb in 2019 averaged across all AirKorea sites. High values are spread throughout South Korea and no site meets the 60 ppb air quality standard. Also shown is the monthly timeseries of ozone for the SMA. Ozone levels are

similar to the rest of South Korea though do not show the 2018 dip. The seasonal maximum is in May-August depending on the year.

Figure 2 shows the annual mean 24-hr average $NO_2$ concentration at the ensemble of AirKorea sites. Concentrations of $NO_2$ in the SMA are generally 10 ppb larger than averaged across South Korea. $NO_2$ concentrations peak in

winter and are minimum in summer, as observed elsewhere in East Asia and mostly driven by longer $NO_x$ lifetime and reduced vertical mixing in winter (Lamsal et al., 2010; Shah et al., 2019; Lin et al., 2019; Kim et al., 2020). There is a decreasing trend over the 2015-2019 period as previously reported by Seo et al. (2021), more so in summer than in winter.

**3 Meteorological Correction of 2015-2019 trends**

The 2015-2019 trends in ozone and $NO_2$ concentrations from Figures 1 and 2 could reflect emission trends but also meteorological variability. Here we use a random forest (RF) non-parametric statistical model (Breiman, 2001; Tong



et al., 2003) to isolate and remove the effect of meteorological variability for the 79 AirKorea sites in the SMA
       (Figure 3). The RF model was constructed using R "normalweatherr" packages
       (https://github.com/skgrange/normalweatherr; Grange et al., 2018). Hourly meteorological data are from two sites
       operated by the Korean Meteorological Administration (KMA) within the SMA (https://data.kma.go.kr/data/grnd).
       The RF model is trained to predict the hourly ozone and $NO_2$ concentrations averaged across the 79 AirKorea sites
       using the meteorological data averaged for the 2 KMA sites as well as time of day, day of year, and a long-term
linear trend (Unix time stamp). Explanatory variables for the RF algorithm are listed in Table 1. Training of the RF
       model was conducted on 70% of the input data and the other 30% were withheld as testing data. The number of
       variables used to grow a tree was set to three, the minimum node-size was five, and the number of trees within a
       forest was set to 300.

Figure 4 compares observed and predicted hourly concentrations of ozone and $NO_2$ for the data withheld from
       training. The RF model shows a strong predictive ability (R = 0.93 for ozone, 0.90 for $NO_2$) with negligible mean
       bias (0.23 ppb for ozone, 0.01 ppb for $NO_2$), and root-mean-square errors (RMSE) of 5.9 ppb for ozone and 6.2 ppb
       for $NO_2$ The model has difficulty in capturing the tails of the distribution, which is a well-recognized problem in RF
       algorithms (Zhang and Lu, 2012; Pendergrass et al, 2021).


       The top predictors in the RF fit for ozone are temperature, day of year, relative humidity, hour of day, and wind
       speed, in that order, consistent with previous studies for urban areas (Sillman and Samson, 1995; Jacob and Winner,
       2009; K. Li et al., 2020). The top predictors for $NO_2$ are wind speed, day of year, temperature, hour of day, and
       surface pressure, again consistent with previous studies (Liu et al., 2020; Richmond-Bryant et al., 2018).


       We use the RF model to remove the effect of meteorological variability in driving the 2015-2019 ozone and $NO_2$
       trends by following the technique outlined in Vu et al. (2019). Meteorological variables for a specific hour and date
       in the input dataset are replaced by randomly selecting weather data over the entire study period (2015-2019) at that
       hour of day but for different day of year within a 4-week period (2 weeks before to 2 weeks after the selected date).
This process is repeated 1000 times, and the resulting 1000 RF predictions of ozone and $NO_2$ for that hour and date
       are then averaged to produce meteorology-corrected concentrations from which we recalculate MDA8 ozone and
       24-h averaged $NO_2$ to infer 2015-2019 emission-driven trends.

**4 Emissions-driven trends in ozone and $NO_2$ concentrations, 2015-2019**


       Figure 5 shows the observed and meteorology-corrected trends of monthly 90[th] percentile MDA8 ozone
       concentrations in the SMA from 2015 to 2019. The observations show peak increase in May but highly variable
       trend from month to month driven in part by interannual meteorological variability. The meteorology-corrected data
       show a much smoother behavior with a broad springtime (March-May) maximum in the increasing ozone trend, and
a decreasing trend in August. Meteorology-corrected ozone is highest in May-June for all years. The 2015-2019




trend in meteorology-corrected $90^{th}$ percentile ozone is 0.7, 1.4, and 0.4 ppb $a^{-1}$ for winter, spring, and autumn. The overall trend for summer is not statistically significant, but the trend for June alone is 0.9 ppb $a^{-1}$. This seasonality of ozone trends in the SMA from 2015 to 2019 is consistent with the 2000-2014 results of Jung et al. (2018), who reported a maximum springtime ozone increase in South Korea and an advancement of the ozone season by 2.1 days

per year. Similar seasonality in the ozone trend has been reported for the North China Plain (K. Li et al., 2021), showing a twofold increase in May ozone exceedances above the 75 ppb standard from 2014 to 2019. Ozone production is most likely to be $NO_x$-limited in summer and VOC-limited in spring and fall (Jacob et al., 1995), thus the seasonality of the trend is consistent with VOC-limited conditions.

Figure 6 is same as Figure 5 but for 24-hr average $NO_2$ concentrations. The observations show a >2% $a^{-1}$ decrease in all months except November-February. The meteorology-corrected data show a consistent 5.6% $a^{-1}$ decrease in March-October, or 22% over the four years, and a consistent but weaker 1.5% $a^{-1}$ decrease in November-February. This is consistent with findings from Bae et al. (2020), who reported a 4.4% $a^{-1}$ decline of annual mean $NO_2$ in the SMA for 2015-2018 using surface and satellite observations. Declining $NO_2$ in the SMA can be attributed to policies

to decrease vehicular $NO_x$ emissions (Kim and Lee, 2018). The weaker decline in winter is consistent with findings from Seo et al. (2021), who found that surface $NO_2$ concentrations in the SMA in 2015-2019 declined by 5.3% $a^{-1}$ during morning commute time for the ozone season, but only by 2.6% $a^{-1}$ for the non-ozone season. The weaker response of $NO_2$ to reduced $NO_x$ emissions in winter could be due to ozone titration by emitted NO, which would take place most systematically at night but also extend to daytime if the ozone supply is weak. We find that in

November-February the decline of $NO_2$ during midday (11:00-15:00 LT) is 2.4% $a^{-1}$, greater than twice that at night (23:00-03:00 LT), consistent with ozone titration. For the GEOS-Chem simulations in the following Sections we will assume a 22% decrease of $NO_x$ emissions from 2015 to 2019.

**5 GEOS-Chem simulation**


We use the GEOS-Chem chemical transport model version 13.3.4 (http://geos-chem.org) to interpret the observed ozone and its 2015-2019 trend in the SMA and more broadly in South Korea, including influences from China and the global background. GEOS-Chem has been applied previously in South Korea to investigate ozone production efficiency (Oak et al., 2019), the factors determining ozone seasonality (Lee and Park, 2022), and the photochemical

environment for ozone production (Yang et al., 2022). Park et al. (2021) previously found that GEOS-Chem version 12.7.2 underestimated free tropospheric ozone over South Korea by 20-30 ppb. Addition of detailed aromatic chemistry in version 13.3.4 (Bates et al., 2021) was subsequently found to increase net ozone production over South Korea by 37% (Oak et al., 2019). Here we also add particulate nitrate photolysis and suppression of sea salt aerosol debromination to the model following Shah et al. (2022), and as we will see this largely corrects the remaining

model ozone bias over East Asia.





We use a nested-grid version of GEOS-Chem driven by MERRA-2 assimilated meteorological data with a horizontal resolution of 0.5x0.625° over East Asia (25°-50° N, 105°-140° E; domain of Figure 7). Chemical boundary conditions at the edges of the nested domain are updated every 3 hours from a global simulation with

4°x5° resolution. We conduct a full-year simulation for 2016 with six months of initialization. Global anthropogenic emissions are from the Community Emissions Data System (CEDS) global inventory (Hoesly et al., 2018) and are superseded with regional emission inventories for South Korea (KORUSv5, http://aisl.konkuk.ac.kr) and China (Multi-resolution Emission Inventory, Zheng et al., 2018). Natural emissions include $NO_x$ from lightning (Murray et al., 2012) and soil (Hudman et al., 2012), MEGANv2 biogenic volatile organic compounds (VOCs) (Guenther et al.,

2012), dust (Meng et al., 2021), and sea salt (Jaeglé et al., 2011). Open-fire emissions are from the Global Fire Emissions Database version 4 (GFED4; van der Werf et al., 2017).

Ships are a relatively large source of $NO_x$ in East Asia. The standard GEOS-Chem model includes pre-processing of ship emissions with the PARAmeterization of emitted $NO_x$ (PARANOX) algorithm (Vinken et al., 2011) to account

for the non-linear chemistry occurring during the dispersion of ship exhaust plumes. PARANOX is a plume-in-grid formulation where ship emissions are aged chemically for 5 hours before being released into the model grid. This greatly reduces the ozone yield from ship $NO_x$ emissions, which would otherwise be diluted by the model in a relatively clean environment where the ozone production efficiency is very high. PARANOX was intended for global model simulations with grid resolution of hundreds of km (Holmes et al., 2014) and its application to higher-

resolution simulations is questionable, particularly over East Asia where the maritime environment is highly polluted (Cuesta et al., 2018; Peterson et al., 2019; Jung et al., 2022). Here we disable PARANOX for the nested simulation and find that this increases ozone over the Yellow Sea in May by 1 ppb on average.

We evaluate our GEOS-Chem simulation for 2016 with MDA8 ozone observations from the AirKorea network in

South Korea and the Ministry of Energy and Environment (MEE) monitoring network in China (http://data.epmap.org/page/index). Observations of seasonal mean 90[th] percentile MDA8 ozone overlaid against our GEOS-Chem simulation are shown in Figure 7. There is good agreement between GEOS-Chem and observations in all seasons, with a spatial correlation coefficient R>0.7 and a mean bias <4 ppb.

We evaluated GEOS-Chem's ability to reproduce the seasonal cycle of ozone in the three megacity clusters of Seoul Metropolitan Area (SMA), Beijing-Tianjin-Hebei (BTH), and Yangtze River Delta (YRD). Figure 8 shows the monthly 90[th] percentile MDA8 ozone for 2016 averaged over all network sites in each cluster. The simulated seasonal cycle is consistent with observations (R > 0.95 and mean bias <6.0 ppb).

May is of particular interest in the SMA because this is when ozone and its increasing trend are highest in the meteorology-corrected data (Figure 5). Previous model comparisons to extensive vertical profiles taken during the KORUS-AQ aircraft campaign over South Korea in May-June 2016 showed large underestimates, with GEOS-Chem version 12.7.2 being too low by 20-30 ppb (Park et al., 2021). The model updates described above largely



correct this underestimate (Yang et al., 2022). Figure 9 compares our simulated GEOS-Chem ozone profile to the
mean of 15 ozonesonde observations over Olympic Park in Seoul taken during the KORUS-AQ campaign on DC-8
flight observation days (15 profiles in total).  Our simulation has a low bias of only 5.4 ppb in the free troposphere.

To investigate and diagnose the ability of GEOS-Chem to reproduce the observed 2015-2019 ozone trend in the
SMA, we performed simulations with 2016 meteorology (January 2016- December 2016) and perturbed emissions
in China and South Korea for 2015 and 2019 to simulate the 2015-2019 trend. The sensitivity simulations used 6
months of initialization. China emissions in 2015 are from MEIC (Zheng et al., 2018) but MEIC does not extend
beyond 2017. Following K. Li et al. (2021), we scaled 2017 MEIC emissions to 2019 based on observed MEE
network trends. Overall, emissions in China declined from 2015 to 2019 by 16% for $NO_x$, 50% for $SO_2$, 23%, for
CO, and 32% for primary $PM_{2.5}$, with flat VOC emissions (K. Li et al., 2021). Anthropogenic emissions for South
Korea in 2015 are taken from the KORUSv5 inventory (http://aisl.konkuk.ac.kr). For 2019 we decrease $NO_x$
emissions in South Korea by 22% (Section 4) and apply no other changes to South Korea emissions, including
VOCs for which emission trends are not clear as mentioned in the Introduction. We also do not apply trends to ship
emissions.

Figure 10 shows the emission-driven trends of 90[th] percentile MDA8 ozone from 2015 to 2019 in the SMA for both
meteorology-corrected observations (data from Figure 5) and GEOS-Chem in individual months. The model trend is
obtained by subtraction of results from simulations with 2015 and 2019 emissions, both using the same 2016
meteorology. GEOS-Chem reproduces the general magnitude and seasonality of the observed trend. It reproduces in
particular the April-May maximum in the trend.

## 6 Attribution of ozone and its 2015-2019 trend over South Korea

We exploit the success of GEOS-Chem in simulating ozone over East Asia and its trend over the SMA to investigate
the causes. We focus on May, where both ozone concentrations and its increasing trend in the meteorology-
corrected data for the SMA are the highest. In addition to the baseline simulation described in Section 5, we also
conduct sensitivity simulations for both emission years to isolate the effects of anthropogenic emissions from South
Korea, China, ships, and East Asia as a whole by zeroing the corresponding emissions including $NO_x$, VOCs, CO,
and $PM_{2.5}$. The same global boundary conditions described above are used for each of these cases, with 6 months of
initialization.

Figure 11 shows the distribution of simulated 90[th] percentile MDA8 ozone for May using 2015 emissions, the
difference when using 2019 emissions, and the contributions from South Korea and China as determined from the
sensitivity simulations with the corresponding emissions shut off. 2015 values in the baseline simulation average
85.8 ppb in the SMA and 90.1 ppb for all of South Korea, and the 2019-2015 difference averages +6.2 ppb for the
SMA while southern parts of the country show decreases. Zeroing out South Korea emissions has remarkably little





effect on SMA concentrations, which remain at 84.1 ppb for 2015, though the 2015-2019 trend is now near zero. Zeroing out China emissions decreases SMA ozone concentrations to 79.8 ppb but the 2015-2019 increase remains at 5.6 ppb. We conclude that the 2015-2019 ozone increase in the SMA can be attributed to the decrease of domestic $NO_x$ emissions under VOC-limited conditions. When emissions from China are zeroed out, we find a 6 ppb ozone

decrease in the SMA and a 8 ppb decrease in South Korea as a whole, representing a significant but relatively modest improvement. The 2015-2019 ozone trend over the SMA is affected by less than 1 ppb, confirming that this trend is mainly driven by domestic emission changes.

A notable result is that ozone levels over South Korea remain very high at about 80 ppb even when emissions from

either South Korea or China are totally shut off. Lee and Park (2022) previously found with GEOS-Chem that surface ozone over South Korea in April hardly changes when domestic emissions are shut off, and here we find that zeroing China emissions also has only a modest effect over South Korea. This resilience is indicative of a major contribution to ozone pollution from the northern mid-latitudes background external to East Asia.

Figure 12 further explores the role of this East Asia background in a simulation with anthropogenic emissions shut off throughout the nested model domain. The 90[th] percentile MDA8 ozone drops to 55 ppb in South Korea, meeting the 60 ppb standard but still extremely high, and indicating that even small anthropogenic emissions would cause ozone to rise above the standard. This high East Asia background affects northern China even more. Lam and Cheung (2022) previously found with GEOS-Chem that the mean MDA8 background ozone over China in April is

53 ppb, and we find here that the 90[th] percentile over northern China reaches 70 ppb. This East Asia background ozone is much higher than the corresponding North American background of 20-40 ppb previously reported in studies of US ozone pollution (Fiore et al., 2003; Zhang et al., 2011; Emery et al., 2012; Jaffe et al. 2018). Such a high East Asia background is reflected in the observation of 75 ppb ozone in the free troposphere (Figure 9) while comparable ozonesonde observations over the western US in spring show mean values of 60 ppb (Zhang et al.,

2014). Satellite observations of free tropospheric ozone also show particularly high values over East Asia (Hu et al., 2017; Gaudel et al., 2018). High free tropospheric ozone over East Asia in spring could reflect regional downwelling from the stratosphere associated with cyclogenesis (Hwang et al., 2007). It could also reflect the observed rise of free tropospheric ozone at northern mid-latitudes and particularly over East Asia in recent decades (Gaudel et al., 2018; Lee et al., 2021; Wang et al., 2022) which could possibly be due to increasing emissions in India and the

Middle East (Anwar et al., 2021; Ding et al., 2022; Anenberg et al., 2022). We find from analysis of sonde observations no significant trend in free tropospheric ozone over South Korea during 2015-2019, meaning that the background is not responsible for the observed increase in surface ozone over that period. Domestic emissions are likely responsible, as discussed above.

Additional panels in Figure 12 show the enhancement of ozone above the East Asian background due to emissions from the Yellow Sea (ships), South Korea, and China. Emissions from ships in the Yellow Sea enhance 90[th] percentile MDA8 ozone over South Korea by only a few ppb, although they can drive ozone concentrations over the





ocean in excess of 90 ppb. Despite ship traffic in the Yellow Sea being intense, the $NO_x$ emissions are still small relative to continental emissions. Emissions from South Korea alone push ozone to almost 80 ppb over South Korea,

with even larger increases over the surrounding oceans reflecting VOC-limited conditions over land. In this way, emissions in South Korea push ozone in the Shandong peninsula in China to over 80 ppb. Emissions in China have a comparable effect on ozone over South Korea as domestic emissions.

**7 Conclusions**


We examined the factors controlling the high and increasing surface ozone concentrations over South Korea and particularly in the Seoul Metropolitan Area (SMA). Ozone in South Korea has risen steadily over the past two decades and is everywhere far in excess of the 60 ppb air quality standard set by the South Korean government in 2015. Improved understanding of the causes of elevated ozone in South Korea is critical for developing effective

emission control strategies.

Analysis of 2015-2019 data from the AirKorea network of air quality monitoring sites shows elevated ozone throughout South Korea, with 90th percentile ozone averaged across all sites exceeding 75 ppb every year and increasing over the period. $NO_2$ concentrations also measured at AirKorea sites are typically >10 ppb higher in the

SMA than elsewhere, with maximum concentrations in winter and a decrease over the 2015-2019 period.

We used a random forest (RF) non-parametric statistical model to isolate and remove the effect of meteorological variability on 2015-2019 ozone and $NO_2$ trends in the SMA. Meteorology-corrected ozone is highest in May-June for all years and increases at the fastest rate of 1.5 ppb $a^{-1}$ in April-May. Meteorology-corrected $NO_2$ is highest

during November-March and lowest in July-August. During the ozone season of March-October, $NO_2$ shows a consistent decline of 5.6% $a^{-1}$ over the 2015-2019 period, whereas in winter the decline is lower at 1.3% $a^{-1}$. The March-October trend in $NO_2$ concentrations suggests that $NO_x$ emissions declined by 22% from 2015 to 2019.

We used the GEOS-Chem chemical transport model to interpret the elevated ozone and its 2015-2019 trend in the

SMA and more broadly in South Korea, including influences from China and the global background. We improved on previous versions of the model, which substantially underestimated tropospheric ozone over South Korea, through the addition of detailed aromatic chemistry in version 13.3.4 (Bates et al., 2021), the removal of sea salt aerosol debromination, and the addition of particulate nitrate photolysis (Shah et al., 2022). The resulting model can reproduce the seasonality and spatial distribution of surface ozone in South Korea and China without significant

bias. It reproduces the high free tropospheric ozone concentrations observed over Seoul during the KORUS-AQ campaign in May-June 2016 (75 ± 7 ppb) with only a 5 ppb low bias. Implementing in the model the 2015-2019 emission decreases in Korea and China reproduces the observed seasonality and magnitude of the meteorology-corrected ozone trend over the SMA.



We went on to use GEOS-Chem sensitivity simulations for emission years 2015 and 2019 to better understand the factors contributing to elevated ozone in the SMA and South Korea, focusing on May when meteorology-corrected ozone and its increase are the highest. We find that the 2015-2019 ozone increase in the SMA can be explained by the 22% decrease of domestic $NO_x$ emissions in South Korea, reflecting the VOC-limited conditions for ozone production.  We also find that emissions in China and South Korea contribute equally to elevated ozone over South

Korea, while ships only contribute a small amount. VOC emission reductions would be expected to decrease ozone in South Korea, but we find that concentrations remain over 80 ppb even if emissions from South Korea or from China are zeroed out. The East Asia background, defined by zeroing out all anthropogenic emissions over East Asia, is very high at 55 ppb, implying that the 60 ppb air quality standard in South Korea is not achievable without addressing the origin of this elevated background.


*Code Availability*

The code used in this work is available upon request.

*Data Availability*

Ground-based measurements from the AirKorea national air quality network of the South Korea Ministry of Environment are available at http://www.airkorea.or.kr/web. Ozonesonde data from the KORUS-AQ data archive are available at https://www-air.larc.nasa.gov (KORUS-AQ Science Team, 2019). Meteorological data from the Korean Meteorological Administration (KMA) are found at https://data.kma.go.kr/data/grnd.

*Author Contribution*

The original draft preparation was done by NKC, with review and editing by DJJ, LHY, SZ, VS, SKG, RMY, SK, and HL. DJJ contributed to project conceptualization. Modeling was done by NKC, with additional support from LHY, SZ, VS, and RMY. The formal analysis was conducted by NKC with additional support from DJJ, LHY, SZ, VS, SKG, and SK.


*Acknowledgments*

This work was funded by the Samsung Advanced Institute of Technology and the Harvard–NUIST Joint Laboratory for Air Quality and Climate (JLAQC). We thank Zongbo Shi and Tuan Vu for their helpful insight on removing the effect of meteorology on pollutant trends.

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



## Maximum monthly 90th percentile MDA8 ozone concentration, ppb

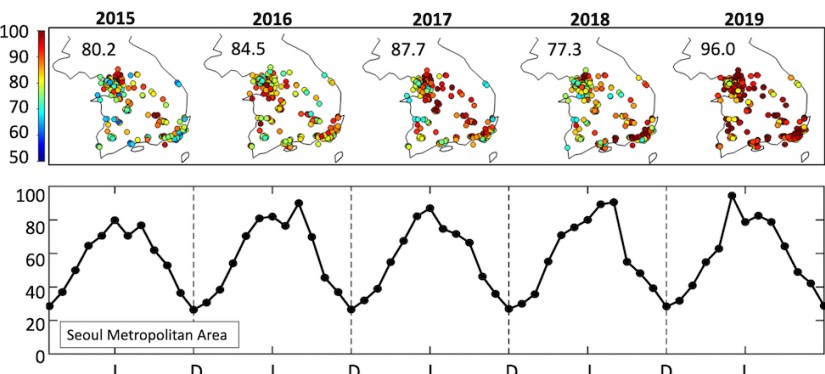

**Figure 1:** 90th percentile maximum daily 8-h average (MDA8) ozone concentrations in South Korea for 2015-2019. The top row shows the maximum monthly 90th percentile ozone at individual AirKorea sites. The mean of this statistic across the ensemble of sites is shown inset. The bottom row shows 90th percentile MDA8 ozone averaged for individual months over sites within the Seoul Metropolitan Area (SMA, 126.7°E–127.3°E, 37.2°N–37.8°N. Tick marks are for June and dashed lines are for December. Only sites with over 90% of observational coverage for the 2015 to 2019 period are included in this analysis.

## 24-h average NO₂ concentration, ppb

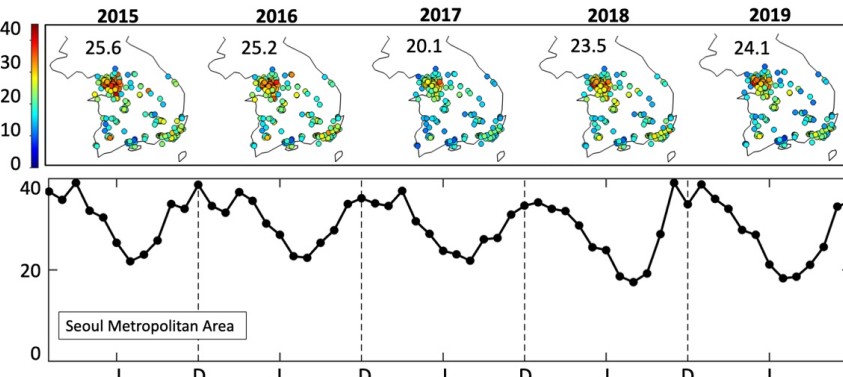

**Figure 2:** Same as Figure 1 but for 24-h average annual mean NO₂ concentrations.



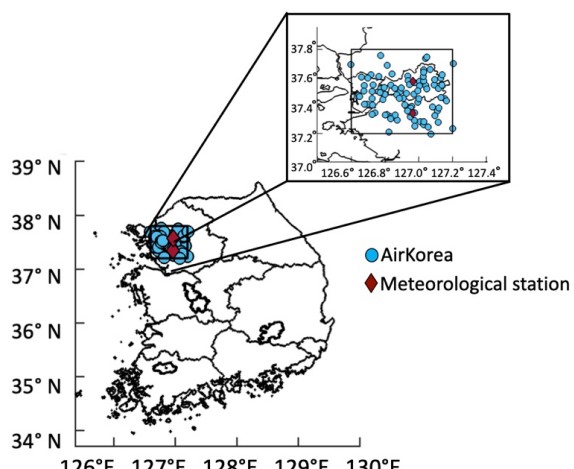

**Figure 3:** AirKorea monitoring sites in the Seoul Meteorological Area (SMA) with hourly ozone and NO$_2$ concentration data for 2015-2019. Red diamonds show the two meteorological sites in the SMA operated by the Korean Meteorological Administration (https://data.kma.go.kr/data/grnd).


**Table 1.** Random forest predictor variables for hourly ozone and NO$_2$ concentrations[a]

| Meteorology[b] | |
|---|---|
| | Wind speed |
| | Wind direction |
| | Temperature |
| | Surface pressure |
| | Relative humidity |
| Time | |
| | Day of Year[c] |
| | Unix time[d] |
| | Hour of day |

[a] Hourly explanatory variables in the random forest (RF) model fitted to hourly ozone and NO$_2$ concentrations averaged across 79 AirKorea sites in the Seoul Metropolitan Area (SMA) for 2015-2019.
[b] Meteorological data are from the two SMA Synoptic Meteorological Observation stations (https://data.kma.go.kr/data/grnd) located at Gwanaksan (126.975°E, 37.345°N) and Seoul (126.980°E, 37.585°N). Data are averaged across the two stations for input to the RF model.
[c] Day of year, used as a seasonal term
[d] Used as a linear trend term



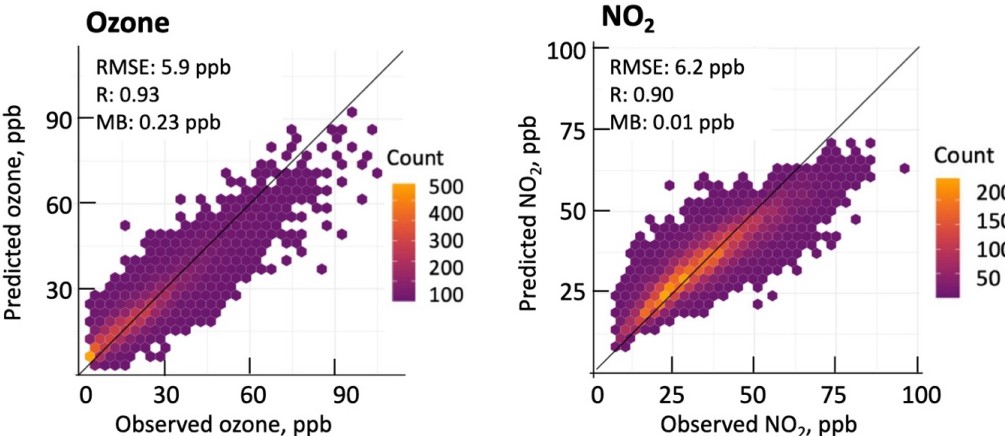


**Figure 4:** Performance of the random forest (RF) model in fitting 2015-2019 hourly ozone and NO₂ concentrations in the Seoul Metropolitan Area (SMA). The RF model is trained on hourly concentrations averaged across 79 AirKorea monitoring sites in the SMA (Figure 3). The Figure compares predicted and observed values for the 30% of data withheld from training. Comparison statistics are shown inset including root-mean-square error (RMSE),
correlation coefficient (R), and mean bias (MB). Also shown are the 1:1 lines. Count refers to the number of data points within a given (ozone, NO₂) data bin (individual symbol).

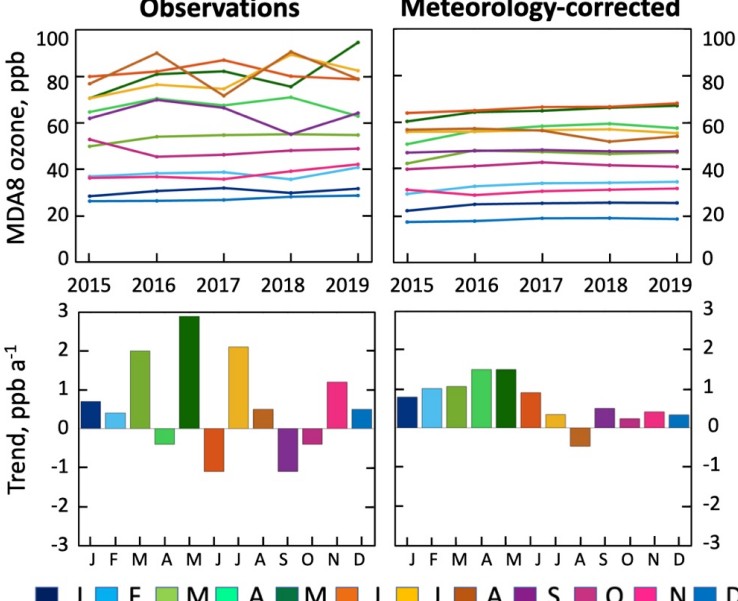

**Figure 5:** 2015-2019 trends in monthly 90th percentile MDA8 ozone averaged across the 79 AirKorea sites in the
Seoul Metropolitan Area (SMA). The left panels show the observed trends for individual months and the right panels show meteorology-corrected trends. The bottom panels show the 2015-2019 slopes for individual months obtained by ordinary least square regressions of the data in the top panels.





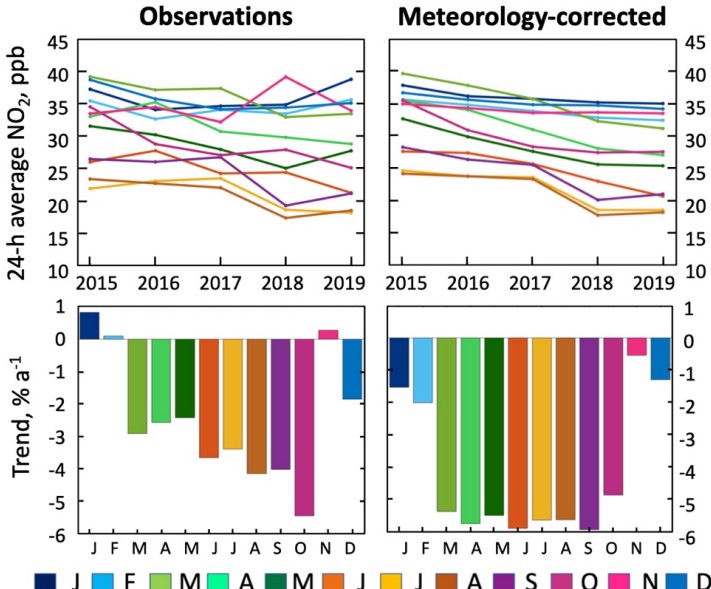

**Figure 6**: Same as Figure 5 but for 24-h average NO₂ concentrations. Trends are shown in % a⁻¹ relative to the 2015-2019 mean.

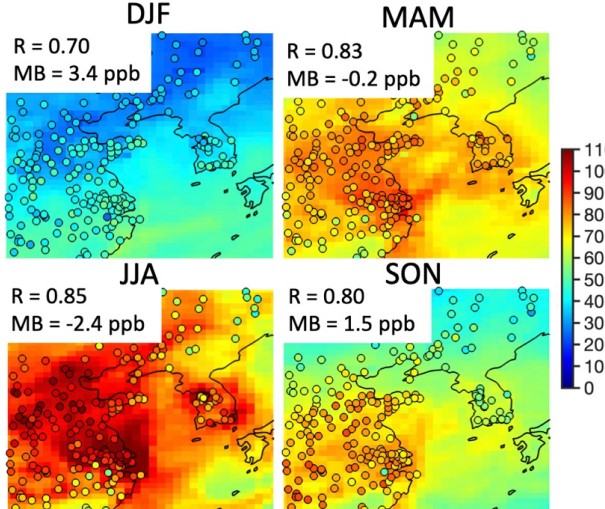

**Figure 7:** Monthly 90[th] percentile MDA8 ozone over South Korea and China for different seasons in 2016. GEOS-Chem model results for each season (background contours) are compared to AirKorea and MEE network observations (symbols). 50% of network sites have been culled randomly for visualization purposes. GEOS-Chem correlation coefficient (R) and mean bias (MB) relative to observations are shown inset.





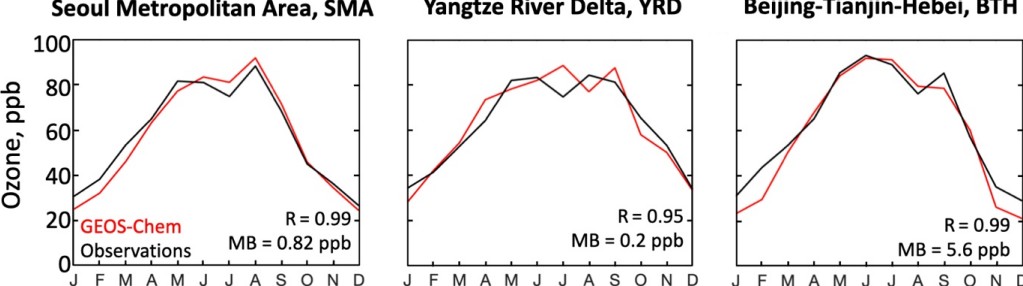

**Figure 8:** Seasonal variation of monthly 90[th] percentile MDA8 ozone in three megacity clusters in 2016. The clusters are the Seoul Metropolitan area (SMA; 126.7°E–127.3°E, 37.2°N–37.8°N), Yangtze River Delta (YRD; 30°–33°N, 118°–122°E), and Beijing-Tianjin-Hebei (BTH; 37°–41°N, 114°–118°E). GEOS-Chem results are compared to observations and the corresponding correlation coefficient (R) and mean bias (MB) are shown inset. The 90[th] percentiles are computed from the time series of spatial mean concentrations for each cluster, with GEOS-Chem sampled at the network sites.

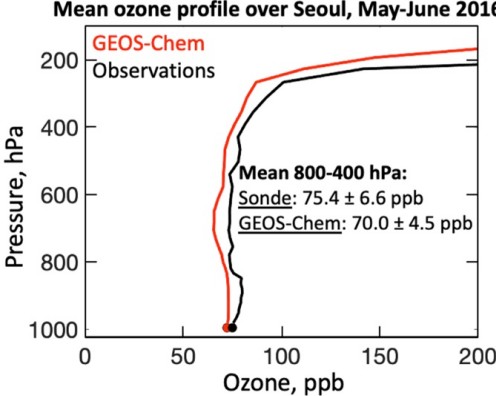

**Figure 9:** Mean vertical ozone profile over Olympic Park (37.522°N, 127.124°E), Seoul during KORUS-AQ (May-June 2016). Observations are from 15 ozonesondes launched at 13:00 local time on KORUS-AQ flight days. GEOS-Chem model results are sampled at the observation times. The circles show the surface ozone concentrations at 13:00 local time in GEOS-Chem and at the AirKorea site in closest proximity.



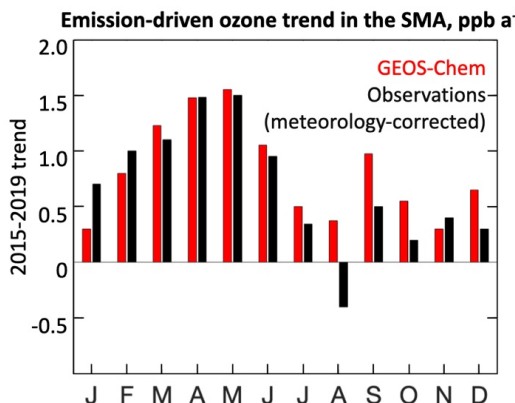

**Figure 10:** Emission-driven trends in 90th percentile MDA8 ozone from 2015 to 2019 in the Seoul Metropolitan Area (SMA) for individual months. The observed meteorology-corrected trend is as shown in Figure 5. The modeled trend is obtained by subtraction of results from simulations with 2015 and 2019 emissions, both using the same 2016 meteorology.

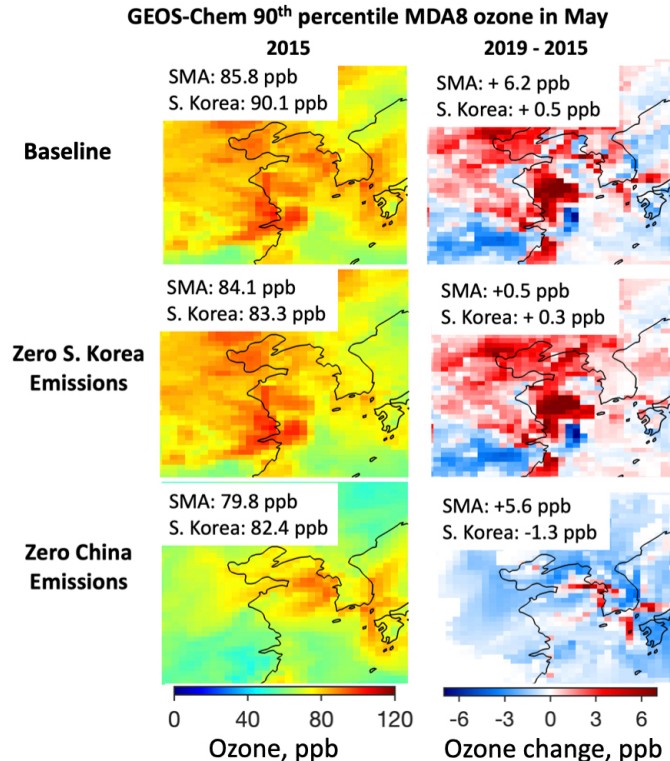


**Figure 11:** Emission-driven ozone changes over East Asia from 2015 to 2019 in GEOS-Chem. Results show the 90th percentile MDA8 ozone for May simulated by GEOS-Chem using 2015 emissions, and the difference using 2019 emissions, both for the same meteorological year. The top row shows the baseline simulation described and evaluated with observations in Section 5. The middle and bottom rows show sensitivity simulations with zero





anthropogenic emissions in South Korea and China respectively. Spatially averaged values for the Seoul Metropolitan Area (SMA) and South Korea are given inset.

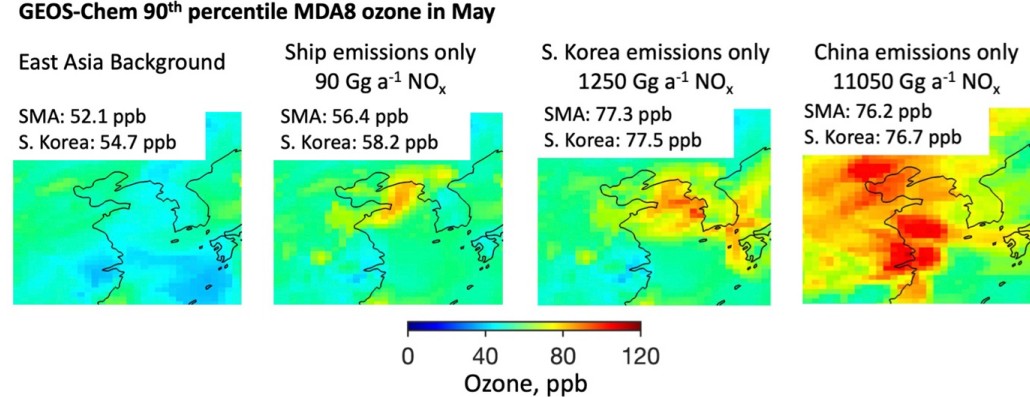

**Figure 12:** East Asia background ozone and individual enhancements due to anthropogenic emissions from ships in
the Yellow Sea (north of 30.5° N), South Korea, and China. Results show the monthly mean 90th percentile MDA8 ozone for May simulated by GEOS-Chem for meteorological year 2016.