# Peer review of "Why is ozone in South Korea and the Seoul Metropolitan Area so high and increasing?"

_EGUsphere, 2022_

## Author Response (AR1)

**Response to comments by Referee # 1**

The manuscript by Colombi et al. (2022) explores the 2015-2019 trends in surface ozone and $NO_2$ concentrations over South Korea and the Seoul Metropolitan Area (SMA) using observations and the GEOS-Chem model. The authors 1) quantified emissions-driven trend in ozone and $NO_2$ concentrations by removing the effect of meteorological variability during the period, 2) successfully simulated surface ozone over South Korea and China by including recent model chemical updates, and 3) identified factors deriving high surface ozone and increasing trends over SMA. The manuscript is well structured and the findings are useful not only to academia but also to policymakers. I support the publication of this manuscript with minor revisions mostly asking for clarification.

We thank the reviewer for their thoughtful and supportive comments. Our response to specific comments is as follows:

**Specific comments**

- **Use of 90th percentile MDA8 as a metric**

As highlighted in the introduction (L45 - 46), there is no monitoring site that complies with the MDA8 national air quality standard in South Korea. It is more urgent to grasp the overall status of MDA8 (mean or median MDA8 as a proxy) or the low concentrations (10th percentile MDA8 as a proxy) rather than high concentrations (90th percentile MDA8 as a proxy) for practical applications. Therefore, I recommend using mean, median, or 10th percentile MDA8 rather than 90th percentile MDA8 as a metric.

We feel that $90^{th}$ percentile MDA8 ozone is the best metric to use because the upper tail of ozone concentrations are most harmful to public health, the main motivator for creating the ozone air quality standard. Additionally, by using $90^{th}$ percentile rather than $95^{th}$ percentile or maximum MDA8 ozone, we filter out many of the highly anomalous pollution episodes that may not contribute to the overall trend. We have added in citations for previous studies that motivate the use of $90^{th}$ percentile ozone.

- **Figure 1 and 2 lower panels**

Adding the same metric for South Korea with different color will be helpful in showing relative ozone pollution in SMA compared to the national average.

We have incorporated this suggestion.

- **Section 3 and 4 titles (L105)**

These sections are for SMA only. Please clarify in the title, too.

We have made this correction.

**Technical corrections**

- L25 − 26: ~find an emission-driven … while $NO_2$ decreases by 22%.

Please rephrase. It is unclear whether the $NO_2$ decrease is from emissions or surface observations.

Changed as suggested.

- L33: ~, we find that SMA ozone would still remain above 80 ppb …

What about other parts of South Korea? Because the analysis of South Korea and SMA are mixed, the focus often gets vague.

Ozone concentrations in both the SMA and South Korea as a whole would remain above 80 ppb. We focus certain statistics on the SMA because this is where we have done the meteorological correction for the ozone and NO2 trends. We have made changes to add more clarity as to which parts of the study pertain to the SMA and which apply to South Korea. Additionally, we changed this line of the abstract to include the statistic for South Korea.

- L43 – 46: In 2015 … continued to increase (NIER, 2020).

MDA8 of 60 ppb is a national air quality standard not limited to SMA.

Corrected.

- L123: ~ for $NO_2$

A period is missing next to $NO_2$.

Corrected, thank you for catching this typo.

- L153: ~thus the seasonality … VOC-limited conditions.

Please rephrase. I don't understand what you mean here.

Added in some words to clarify.

**Response to comments by Referee # 2**

The paper analyzes five years (2015-2019) of primarily surface ozone data in South Korea and the Seoul Metropolitan Area (SMA) to determine the cause for recent observed increases in ozone pollution (particularly in spring) to far above the levels of the South Korean 60 ppb ozone standard. The authors report 90[th] percentile MDA8 ozone and daily averaged $NO_2$, showing increases and decreases, respectively, over the 5-year period. Meteorological variability and its effects on surface ozone and $NO_2$ are accounted for using a random forest model, which strengthens the broad conclusions drawn from the raw ozone and $NO_2$ observations alone.

The GEOS-Chem model is then employed to perform sensitivity tests by altering anthropogenic emissions, which indicate that South Korean domestic emissions decreases are the primary driver for recent surface ozone increases (focus is on May) in a VOC-limited regime. The authors find that even a complete zeroing-out of regional emissions leaves a 55 ppb MDA8 ozone background, meaning the 60 ppb South Korean ozone standard is nearly impossible to achieve under current East Asian background ozone conditions.

This is a well-organized, written, and executed study, and its conclusions are clearly stated and understood. I have a few General Comments where I will suggest some topics that need more detail and clarity, and possible additional supporting analyses, but overall these should amount to Minor Revisions after which I recommend this paper for publication in EGUsphere.

We thank the reviewer for their thorough review of the manuscript and insightful comments. Our response to specific comments is as follows:

**General Comments:**

On use of "trend": I'm not sure I would use the word "trend" to describe 5 years of ozone data/results. It is probably more accurate and appropriate to say "tendency" when describing only 2015-2019. The Gaudel et al. (2018) study that you cite (see Fig 14b) indicates that your results are likely a continuation of what has been observed since the early 2000s. The persistence of what are now multidecadal surface ozone increases in South Korea should be discussed in more detail in the Conclusions.

Problem with 'tendency' is that it is often used to denote (P-L). We tried to emphasize in the paper that ozone in South Korea has risen steadily over the past two decades and cite the appropriate literature. We have added a sentence to the conclusion to emphasize that what we find is a continuation of this multidecadal trend.

MDA8 90th Percentile: What is the rationale for using the 90th percentile? Is that what the 60 ppb MDA8 ozone standard in South Korea is based on? If so, that's a simple explanation that should be included in the Introduction. For example, the US 70 ppb standard is based on a three-year average of a station's annual 4th-highest MDA8. Otherwise, alternate explanations for use of the 90th percentile should be stated.

We chose to focus on the 90th percentile MDA8 ozone metric because the upper tail of ozone concentrations are most harmful to public health. We have added a sentence in the introduction to clarify this, as well as citations for previous studies that motivate the use of 90th percentile ozone.

Emissions vs. Surface $NO_2$ Changes: It should be made clear that actual emissions estimates were not computed in this study. Conclusions on lines 332-333 make it seem like you've computed $NO_x$ emissions and their change over 2015-2019.

This was indeed confusing. We have added some words in the conclusion to clarify this.

Free-Tropospheric Ozone Trends: More information is needed for Lines 285-287. Did you perform a trends analysis on ozonesonde data over South Korea (e.g., Pohang; https://woudc.org/archive/Archive-NewFormat/OzoneSonde_1.0_1/stn332/ecc/), or are you referring to other studies? Also, I suggest quadrupling your sample size for the GEOS-Chem KORUS-AQ analysis by including the 42 Taehwa Research Forest ozonesondes (https://www-air.larc.nasa.gov/cgi-bin/ArcView/korusaq?SONDES=1). That location is just on the edge of your SMA domain, and often observed more ozone near the surface than the $NO_x$-heavy Olympic Park location. Demonstration of GEOS-Chem's ability to reproduce observed ozone levels from the surface to free-troposphere over more than one SMA location would greatly increase confidence in your latter results.

Yes, we performed trend analysis for 2015-2019 for sondes at the Pohang, Hong Kong, and Tateno stations. We have added some words to clarify this. Additionally, we have updated Figure 9 to include both Olympic Park and Taehwa observations.

**Specific and Line-by-Line Comments:**

Line 65: Suggest you also cite Sullivan et al. (2019; ACP; https://acp.copernicus.org/articles/19/5051/2019/) which showed ozone lidar and ozonesonde data from KORUS-AQ and frequent high-ozone episodes at the Taehwa Research Forest. See also my comments about leveraging the Taehwa sondes for the GEOS-Chem analysis.

This has been done.

Several Figures other than Fig 3: A small dashed box, for example like in Figure 3, around your defined SMA region would help orient the reader (and me)

Yes, it makes sense that this would be hard for the reader to distinguish the SMA. We have updated Figures 11 and 12 to include a box around the SMA. For figures where the AirKorea network data is plotted, we refrained from adding this box because it makes it difficult to see the observations.

Line 115: Was solar radiation/cloud cover a possible explanatory variable? Seems like that would be important.

We did not include solar radiation as an explanatory variable, but we believe temperature would act as a proxy here. Since the model performed well using the discussed meteorological variables, we did not want to add in additional variables in order to prevent overfitting.

Lines 116-118: I can find specific information on the RF model from cited sources, but at least a cursory explanation on how the model works should be included here.

We have added a sentence to briefly describe the model.

Figures 7-9: As in Figure 4, r, MB, and RMSE should be provided for all figures to provide a complete picture of GEOS-Chem model performance vs. observations.

For Figure 8, we have replaced R with RMSE since this is a better metric to use and the correlation of the observations with GEOS-Chem is obvious from the plot.

For Figure 7, we feel that R is a better metric to use here since we want to emphasize that GEOS-Chem captures the seasonal spatial variability of surface ozone. Adding additional statistics would make the figure look cluttered.

For Figure 9, we feel that including the mean ozone for 800-400 hPa and standard deviation is the best choice for emphasizing the very high observed ozone in the free troposphere and its day to day variability. Adding additional statistics would make the figure look cluttered.

Figure 11: I'll admit it took me an embarrassingly long time to orient myself looking at this map. A box around SMA will help guide the eye. The land/water distinction and South Korea on Figure 7 were easier to immediately pick out because the surface ozone stations are plotted.

Corrected.

Lines 259-261: Make it clearer that these changes refer to the 2015 ozone values, and not the 2015 to 2019 changes in ozone. The discussion of 2015 values and 2019-2015 changes are mixed in this paragraph, sometimes confusingly.

Corrected.

Lines 260-261: Sure, it still doesn't get them close to achieving the 60 ppb standard, but I respectfully disagree that 6 and 8 ppb decreases in MDA8 are modest. Also "significant but relatively modest" seems somewhat mutually exclusive.

We agree and have corrected this.

Line 330: Suggest a rewrite of "We went on to"

Corrected.